# Transformers meet Stochastic Block Models: Attention with Data-Adaptive Sparsity and Cost

**Sungjun Cho**[1]    **Seonwoo Min**[1]    **Jinwoo Kim**[2]
**Moontae Lee**[1,3]    **Honglak Lee**[1]    **Seunghoon Hong**[2,1]
[1]LG AI Research    [2]KAIST    [3]University of Illinois Chicago

## Abstract

To overcome the quadratic cost of self-attention, recent works have proposed various sparse attention modules, most of which fall under one of two groups: 1) sparse attention under a hand-crafted patterns and 2) full attention followed by a sparse variant of softmax such as $\alpha$-entmax. Unfortunately, the first group lacks adaptability to data while the second still requires quadratic cost in training. In this work, we propose SBM-Transformer, a model that resolves both problems by endowing each attention head with a mixed-membership Stochastic Block Model (SBM). Then, each attention head data-adaptively samples a bipartite graph, the adjacency of which is used as an attention mask for each input. During backpropagation, a straight-through estimator is used to flow gradients beyond the discrete sampling step and adjust the probabilities of sampled edges based on the predictive loss. The forward and backward cost are thus linear to the number of edges, which each attention head can also choose flexibly based on the input. By assessing the distribution of graphs, we theoretically show that SBM-Transformer is a universal approximator for arbitrary sequence-to-sequence functions in expectation. Empirical evaluations under the LRA and GLUE benchmarks demonstrate that our model outperforms previous efficient variants as well as the original Transformer with full attention. Our implementation can be found in `https://github.com/sc782/SBM-Transformer`.

## 1 Introduction

The Transformer [38] architecture has been the go-to method for encoding sequential data, due to its superior performance in various tasks such as machine translation [28], image classification [14], and protein language modeling [32]. Its key strength stems from the multi-head attention module, where a so-called *attention score* matrix computes how contextually important one token is to another for all possible token pairs. Each Transformer layer simultaneously pools the token representations based on the attention scores, eventually returning contextualized features without sequentially traversing through the input sequence as its recurrent neural network-based predecessors [16].

A well-known drawback of the original Transformer is its high computational cost in time and memory that increases quadratically with sequence length. This is due to the full pairwise computation of attention scores, which prohibits applying it in tasks involving long-range dependencies such as document summarization [17] or high-resolution image processing [48]. Many works have thus focused on developing more efficient alternatives by exploiting fixed or learnable attention sparsity patterns [8, 46, 20, 12], low-rank approximations [40, 43], or kernelized attention modules [19, 9].

Even though the efficient alternatives hold theoretical expressibility guarantees [45], they are far from sufficient, still failing to convince practitioners to replace the original Transformer. We believe this is mostly due to their lack of adaptability. They apply the same modifications to unanimously sparsify all the attention modules across layers, without considering the tasks at hand. Such strategy

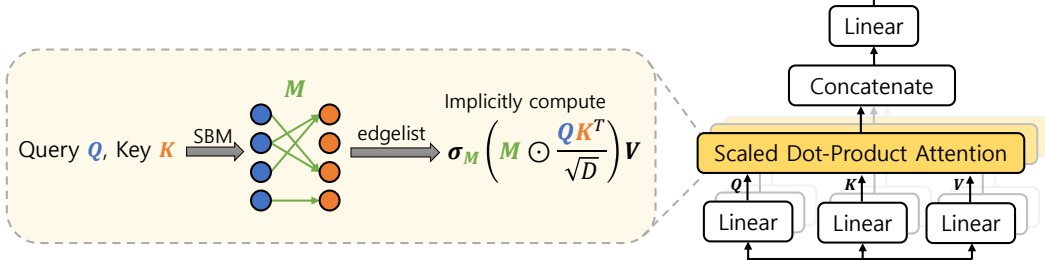

Figure 1: The attention module in SBM-Transformer. In multi-head attention, each attention head samples a bipartite graph connecting queries to keys from an underlying SBM. The adjacency of the sampled graph is used as an attention mask to compute the dot products only for the sampled edges.

imposes inductive bias too strongly and often leads to sub-optimal cost vs. performance trade-offs in downstream tasks [27]. In this work, we argue that to retain the utmost potential of Transformers, each attention module should have the ability to flexibly choose between sparse and full attention. This is especially evident when considering many state-of-the-art systems suggest the need for a mixture of dense and sparse attention layers. For example, a qualitative analysis on pretrained BERT showed that lower layers exhibit broad dense attention while upper layers perform focused sparse attention [10]. In the case of GPT-3 [6], the Transformer blocks are manually arranged to alternate between dense and sparse attention.

To contribute to the efficient Transformers lineage, we propose SBM-Transformer, capable of adjusting its attention sparsity data-adaptively based without fully computing the attention score matrix (Figure 1). Leveraging a mixed-membership Stochastic Block Model (SBM) [2], each attention head samples a bipartite graph connecting queries to keys. Then, the adjacency of the sampled graph is used as an attention mask so that only attention scores corresponding to sampled edges are computed. The overall computational cost is linear in the number of edges, which can range from linear to quadratic in sequence length depending on the data and task under concern. Each attention head is equipped with its own underlying SBM, enabling the model to diversify the attention sparsity across heads and layers. By incorporating a straight-through estimator [4] in the discrete graph-sampling step, SBM-Transformer enjoys end-to-end differentiability and can find the proper attention sparsity based solely upon minimizing the predictive loss. The model can also easily be further regularized by penalizing the number of sampled edges, which results in a lighter model using less computational resources during inference. To the best of our knowledge, our method is the first Transformer architecture that can data-adaptively choose between linear to full attention with respective computational costs. To summarize, our main contributions are as follows:

- We present SBM-Transformer, a novel Transformer of which each attention head can adaptively adjust its attention sparsity as well as computational cost based on the input data.
- To demonstrate the benefit of this flexibility, we theoretically prove that SBM-Transformer retains universal approximability, and also stress-test the model under a synthetic task where full attention is required to achieve 100% accuracy.
- Evaluations on LRA and GLUE benchmarks show that SBM-Transformer outperforms previous efficient Transformer models as well as the vanilla Transformer with dense attention.

## 2 Related Work

In this section we discuss previous efficient Transformer variants and several works similar to ours with respect to adaptively learning sparse attention patterns. We also review several works on SBMs.

**Efficient Transformers.** Many efficient Transformers tackle to reduce the quadratic cost of multi-head attention with different approaches. While we discuss only a handful of representative approaches, a much more comprehensive survey can be found in [37]. The Linear Transformer [19] achieves linear complexity by replacing the softmax with a low-rank kernelized function. Linformer [40] and Nyströmformer [43] use a similar approach by low-rank approximating the attention score matrix. Performer [9] uses positive orthogonal random features to approximate the softmax kernel. Reformer [20] gathers similar tokens together through locality-sensitive hashing (LSH) and performs attention amongst tokens within the same bucket. Of all methods above, our method is

most similar to Reformer, in the sense that we adaptively assign queries and keys into clusters and form a low-rank sparse attention pattern. However, our method performs soft-clustering with much less structural constraints, allowing each attention head to represent a wider variety of dependency structure and to adjust its sparsity towards full attention if needed.

**Adaptive Sparsity.** With respect to flexible training between sparse and dense attention, there exist some works that parameterize how sparse the attention pattern should be based on the input. The Adaptive Sparse Transformer [11] proposed replacing the usual softmax activation with $\alpha$-entmax, in which the $\alpha$ parameter can be differentiably trained to adjust the activation between softmax and sparsemax activation [25]. SparseBERT [34] uses a differentiable masking technique where each attention mask is sampled from a Gumbel-sigmoid distribution using data-independent mask probability parameters. While these methods possess the flexibility to adjust between sparse and full attention based on data, they still require full computation of the attention score matrix before sparsification, and hence are unable to leverage the learned sparsity towards better model efficiency. To the best of our knowledge, ours is the first work to be able to adaptively tune its attention sparsity between sparse to full attention without requiring the explicit computation of the attention score matrix, thereby avoiding quadratic cost when possible.

**Stochastic Block Models.** The Stochastic Block Model (SBM) is a generative model that encodes the latent structure of graphs by grouping nodes into clusters. By modeling the cluster-membership of each node as well as inter-cluster relationships, SBMs can represent a wide variety of graph structures, which is a feature especially useful for generating new graphs or predicting missing edges in noisy data [1]. The standard SBM assigns each node to a single cluster, and the probability of an edge between two nodes strictly depends on the corresponding clusters. Several structural extensions include overlapping SBM [22] and mixed-membership SBM [2], which allow each node to be assigned to multiple clusters. The underlying SBM used by our framework mostly resembles these two variants, while the edge probability is modeled by a nonlinear function of two node embeddings rather than a bilinear one. There exist many other extensions including degree-corrected SBM [18] for multi-graphs and hierarchical SBM [29] for multiplex-graphs. Further details can be found in a recent survey [15].

# 3 Preliminaries: Sparse Transformers

We first introduce the full attention mechanism used in the original Transformer [38] as well as masked attention which will serve as a backbone of our approach.

## 3.1 Full Attention

In vanilla Transformer [38], each attention head takes a sequence of token features as input $\boldsymbol{X} \in \mathbb{R}^{n \times d}$ where $n$ is the sequence length and $d$ the embedding dimension. Weight parameters $\boldsymbol{W}^Q, \boldsymbol{W}^K \in \mathbb{R}^{d \times d_h}$ and $\boldsymbol{W}^V \in \mathbb{R}^{d \times d_h}$ with head-dimension $d_h$ first maps the input features $\boldsymbol{X}$ into query $\boldsymbol{Q}$, key $\boldsymbol{K}$, and value $\boldsymbol{V}$, respectively. Then, the *attention score matrix* is computed with scaled dot-product of queries and keys followed by row-wise softmax activation $\sigma(\cdot)$. Note that explicit computation of this matrix is the main bottleneck of full attention, incurring $\mathcal{O}(n^2)$ asymptotic cost in both time and memory. The value features $\boldsymbol{V}$ are then pooled based on the attention scores, returning the output token representations. Altogether, the operation performed by each attention head can be written as

$$\boldsymbol{Q} = \boldsymbol{X}\boldsymbol{W}^Q, \;\; \boldsymbol{K} = \boldsymbol{X}\boldsymbol{W}^K, \;\; \boldsymbol{V} = \boldsymbol{X}\boldsymbol{W}^V \tag{1}$$

$$\texttt{Attn}(\boldsymbol{X}) = \sigma\left(\frac{\boldsymbol{Q}\boldsymbol{K}^T}{\sqrt{d_h}}\right)\boldsymbol{V}. \tag{2}$$

## 3.2 Masked Attention

One way to remove the quadratic bottleneck from the attention score matrix is to apply a binary mask $\boldsymbol{M} \in \{0,1\}^{n \times n}$ and compute the scaled dot-products $\boldsymbol{Q}_i \boldsymbol{K}_j^T / \sqrt{d_h}$ only if $\boldsymbol{M}_{ij} = 1$. In presence of an attention mask, the operation is modified to

$$\texttt{Attn}_{\text{mask}}(\boldsymbol{X}, \boldsymbol{M}) = \sigma_{\boldsymbol{M}}\left(\boldsymbol{M} \odot \frac{\boldsymbol{Q}\boldsymbol{K}^T}{\sqrt{d_h}}\right)\boldsymbol{V} \tag{3}$$

$$\sigma_{\boldsymbol{M}}(\boldsymbol{A})_{ij} := \begin{cases} \dfrac{\exp(\boldsymbol{A}_{ij})}{\sum_{k \in \{k' | \boldsymbol{M}_{ik'}=1\}} \exp(\boldsymbol{A}_{ik})} & \text{if } \boldsymbol{M}_{ij} = 1 \\ 0 & \text{otherwise} \end{cases} \tag{4}$$

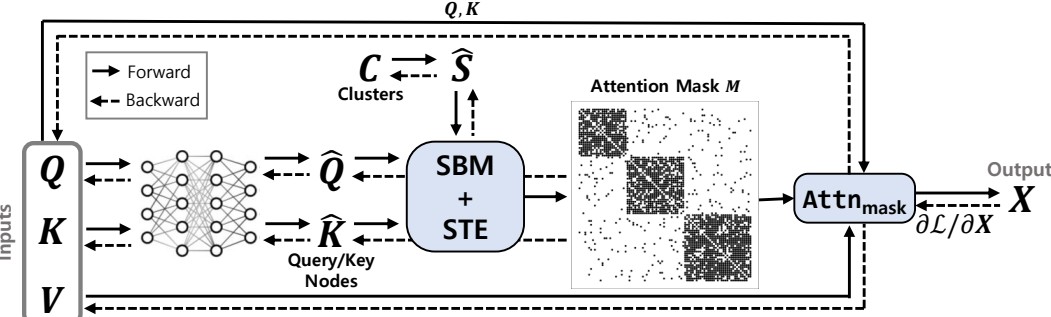

Figure 2: An illustration of the attention mechanism in SBM-Transformer. Each head first maps queries and keys to the node representation space through a shared MLP. The graph sampling module samples an attention mask from a Stochastic Block Model (SBM) parameterized by the node and cluster embeddings. The discrete sampling step is differentiable via a Straight-Through Estimator (STE). Given the mask, the output is computed via masked attention.

where $\odot$ indicates entry-wise multiplication. Note that the masked-softmax $\sigma_{\boldsymbol{M}}(\cdot)$ operator only computes unmasked terms, ensuring that each $(i,j)$-th attention score survives as nonzero if and only if $\boldsymbol{M}_{ij} = 1$. This is thus equivalent to filling in the $(i,j)$-th attention score with $-\infty$ if $\boldsymbol{M}_{ij} = 0$, then applying the standard softmax operator. Most sparsity-based efficient Transformers fall under this formulation, while using different methods to either manually fix or learn the mask $\boldsymbol{M}$. For instance, local attention [8, 3, 46] with a sliding window sets $\boldsymbol{M}_{ij} = 1$ if $|i - j| < c$ for some context window size $c$ while Reformer [20] sets $\boldsymbol{M}_{ij} = 1$ if $\boldsymbol{Q}_i$ and $\boldsymbol{K}_j$ are hashed into the same bucket.

## 4 Our Method: SBM-Transformer

Here we discuss the details of SBM-Transformer (Figure 2). We first illustrate the forward step of our attention module and how the underlying SBM [2] of each head, from which we sample our attention masks, is parameterized by the input tensors. We then discuss how the model enables end-to-end differentiability despite the discrete graph sampling step.

### 4.1 Forward step with the Stochastic Block Model

In our framework, we view the attention mask $\boldsymbol{M}$ as an adjacency matrix of a bipartite graph that connects queries to keys, and let each attention head sample an adjacency matrix that best represents the contextual dependencies amongst input tokens. In order to efficiently sample adjacency matrices while avoiding the quadratic cost, the distribution of graphs must first be parameterized with a sub-quadratic number of latent variables. Stochastic Block Models fit perfectly for our purpose as it models graphs that are low-rank structured with $k$ latent clusters, allowing full parameterization using $\mathcal{O}(nk)$ memory. More concretely, the SBM distribution is defined by two nonnegative node-to-cluster memberships $\boldsymbol{Y}, \boldsymbol{Z} \in \mathbb{R}_+^{n \times k}$ and a so-called block matrix $\boldsymbol{B} \in \mathbb{R}_+^{k \times k}$ that stores the inter-cluster connection probabilities. The probability of node $i$ being connected to node $j$ is computed as $p(i,j) = \boldsymbol{Y}_i \boldsymbol{B} \boldsymbol{Z}_j^T$. Equivalently, the expectation of the adjacency matrix sampled from $\boldsymbol{A} \sim SBM(\boldsymbol{Y}, \boldsymbol{B}, \boldsymbol{Z})$ can be written as $\mathbb{E}[\boldsymbol{A}] = \boldsymbol{Y} \boldsymbol{B} \boldsymbol{Z}^T$.

For proper parameterization of the SBM, we must infer the nonnegative node-memberships and block matrix from the queries and keys. To do so, we equip each attention head a 2-layer $\text{MLP}_{d_h \to d_h}$ with ReLU activation, and a set of $k$ trainable cluster-embeddings $\boldsymbol{C} \in \mathbb{R}^{k \times d_h}$. First, our model computes the block matrix $\hat{\boldsymbol{S}} \in \mathbb{R}_+^{k \times k}$ by taking dot products amongst cluster-embeddings $\boldsymbol{C}$ followed by a 2-dimensional softmax activation. The node embeddings are obtained by processing each query and key through the $\text{MLP}_{d_h \to d_h}$, mapping token representations into the node representation space. The memberships of query and key nodes, which we denote by $\hat{\boldsymbol{Q}}$ and $\hat{\boldsymbol{K}}$, are then inferred by taking dot products of node and cluster embeddings, followed by a sigmoid function. The block matrix $\hat{\boldsymbol{S}}$, query node-memberships $\hat{\boldsymbol{Q}}$, and key node-memberships $\hat{\boldsymbol{K}}$ altogether provide a well-defined parameterization for the SBM. Thus, a bipartite graph adjacency $\boldsymbol{M} \in \{0, 1\}^{n \times m}$ can be sampled from $\boldsymbol{M} \sim SBM(\hat{\boldsymbol{Q}}, \hat{\boldsymbol{S}}, \hat{\boldsymbol{K}})$ with expectation $\mathbb{E}[\boldsymbol{M}] = \hat{\boldsymbol{Q}} \hat{\boldsymbol{S}} \hat{\boldsymbol{K}}^T$: the probability of connecting query $\boldsymbol{Q}_i$ to key $\boldsymbol{K}_j$ equals $p(i,j) = \hat{\boldsymbol{Q}}_i \hat{\boldsymbol{S}} \hat{\boldsymbol{K}}_j^T$. Formally, the sampling procedure can be written as

**Algorithm 1:** fastRG$(\boldsymbol{Y}, \boldsymbol{B}, \boldsymbol{Z})$[33]

---

**Input** : $\boldsymbol{Y} \in \mathbb{R}_+^{n \times k}$, $\boldsymbol{B} \in \mathbb{R}_+^{k \times k}$, $\boldsymbol{Z} \in \mathbb{R}_+^{n \times k}$
**Output**: $\boldsymbol{M} \in \{0,1\}^{n \times n}$ with $\mathbb{E}[\boldsymbol{M}] = \boldsymbol{Y}\boldsymbol{B}\boldsymbol{Z}^T$

1 Compute diagonal matrices $\boldsymbol{D_Y} = (\mathrm{diag}(\mathbf{1}\boldsymbol{Y}))^{-1}$ and $\boldsymbol{D_Z} = (\mathrm{diag}(\mathbf{1}\boldsymbol{Z}))^{-1}$
2 Column-normalize $\overline{\boldsymbol{Y}} = \boldsymbol{Y}\boldsymbol{D_Y^{-1}}$ and $\overline{\boldsymbol{Z}} = \boldsymbol{Z}\boldsymbol{D_Z^{-1}}$
3 Compute $\overline{\boldsymbol{B}} = \boldsymbol{D_Y}\boldsymbol{B}\boldsymbol{D_Z}$
4 Sample number of edges $m \sim \mathrm{Poisson}(\mathbf{1}\overline{\boldsymbol{B}}\mathbf{1}^T)$
5 Initialize $\boldsymbol{M} = \mathbf{0}$
6 **for** $i = 1 : m$ **do**
7 $\quad$ Sample $(U, V)$ from $\{1, \ldots, k\} \times \{1, \ldots, k\}$ with $Pr(U = u, V = v) \propto \overline{\boldsymbol{B}}_{uv}$
8 $\quad$ Sample source $I$ from $\{1, \ldots, n\}$ with $Pr(I = i) = \overline{\boldsymbol{Y}}_{iU}$.
9 $\quad$ Sample destination $J$ from $\{1, \ldots, n\}$ with $Pr(J = j) = \overline{\boldsymbol{Z}}_{jV}$
10 $\quad$ Set $\boldsymbol{M}_{IJ} = 1$.
11 **end**

$$\hat{\boldsymbol{S}} = \texttt{softmax}(\boldsymbol{C}\boldsymbol{C}^T) \tag{5}$$

$$\hat{\boldsymbol{Q}} = \texttt{sigmoid}(\mathrm{MLP}_{d_h \to d_h}(\boldsymbol{Q})\boldsymbol{C}^T) \tag{6}$$

$$\hat{\boldsymbol{K}} = \texttt{sigmoid}(\mathrm{MLP}_{d_h \to d_h}(\boldsymbol{K})\boldsymbol{C}^T) \tag{7}$$

$$\boldsymbol{M} \sim SBM(\hat{\boldsymbol{Q}}, \hat{\boldsymbol{S}}, \hat{\boldsymbol{K}}) \tag{8}$$

For the last sampling step, we incorporate a fast random graph sampling algorithm fastRG (Alg. 1, [33]) that can sample graphs from a SBM in time and memory asymptotically linear in the number of edges. One advantage of fastRG is that each edge can be sampled in parallel, allowing high efficiency with the help of multiprocessing. A more significant feature of the method is that the number of edges, which determines the overall cost, is sampled from a Poisson distribution with input-dependent mean (Line 4). Thus, the model can dynamically adjust its computational cost between linear and quadratic in sequence length based on the data.

Figure 3 shows example placements of nodes and clusters on the $d_h$-dimensional space to show how the sparse structure is determined. If all nodes and clusters are gathered closely, then all entries in $\hat{\boldsymbol{Q}}$ and $\hat{\boldsymbol{K}}$ become close to 1, resulting in $p(i, j) \approx 1$ for all $i, j$ and hence a dense $\boldsymbol{M}$. If clusters are well-separated but each surrounded by some set of nodes, $\hat{\boldsymbol{S}}$ becomes close to diagonal while each row in $\hat{\boldsymbol{Q}}$ and $\hat{\boldsymbol{K}}$ is close to a one-hot vector indicating the cluster nearby. Such setting leads to a block diagonal mask similar to LSH bucketing of Reformer [20]. Lastly, if all clusters are far apart from the nodes, both $\hat{\boldsymbol{Q}}$ and $\hat{\boldsymbol{K}}$ approximately equal zero, zeroing out all the edge probabilities.

## 4.2 Backward Step with Straight-Through Estimator

The graph sampling procedure is naturally a discrete operation. Thus, naive backpropagation cannot learn the proper parameterization for the SBM that minimizes the predictive loss. To cope with this non-differentiability, we incorporate a Straight-Through Estimator (STE) [4] to pass the gradient beyond the discrete sampling step. The STE enables providing the gradient $\partial\mathcal{L}/\partial\boldsymbol{M}_{ij}$ to the probability for each sampled edge $(i, j)$ (Eqn. 9). It works as if we had used a continuous mask $\boldsymbol{M} \odot \mathbb{E}[\boldsymbol{M}]$ that stores the probability of each sampled edge instead of the binary mask $\boldsymbol{M}$ during forward propagation. This way, the probabilities of sampled edges can be learned end-to-end: the gradients provide information on whether each sampled edge was useful or not for prediction.

$$\frac{\partial\mathcal{L}}{\partial p_{ij}} := \frac{\partial\mathcal{L}}{\partial\boldsymbol{M}_{ij}} = \begin{cases} \dfrac{\partial\mathcal{L}}{\partial\boldsymbol{A}_{ij}} \cdot \dfrac{\boldsymbol{Q}_i\boldsymbol{K}_j^T}{\sqrt{d_h}} & \text{if } \boldsymbol{M}_{ij} = 1 \\ 0 & \text{otherwise} \end{cases} \quad \text{where } \boldsymbol{A} := \boldsymbol{M} \odot \dfrac{\boldsymbol{Q}\boldsymbol{K}^T}{\sqrt{d_h}} \tag{9}$$

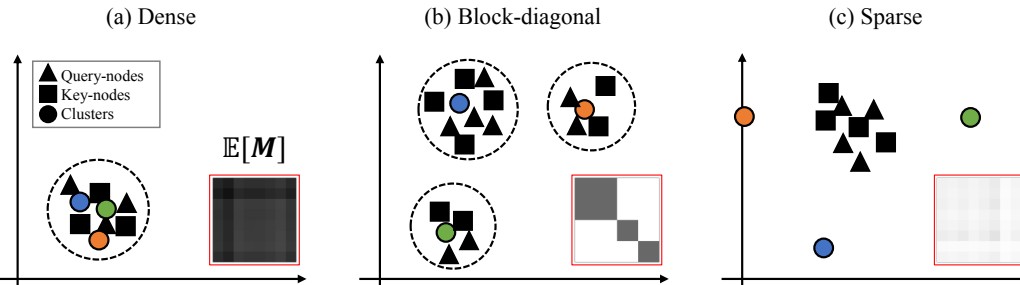

Figure 3: Representative examples from the SBM and resulting mask expectations (darker grid indicates edge probability closer to 1). (a) The expected mask is dense if all nodes and clusters are collapsed within a small region. (b) Clear-cut groups in the embedding space induce a block-diagonal mask. (c) Clusters located far apart from nodes lead to sparse masks.

**Random Edge Exploration.**   While this approach enables backpropagation in the same $\mathcal{O}(m)$ cost as in the forward step, this comes at the expense of not being able to propagate information through edges that were not sampled. This can be problematic when an edge probability accidentally collapses to zero, after which the edge becomes unlikely to ever be sampled even when it may be useful for the prediction task at hand. Therefore, we add a small perturbation $\delta > 0$ to each edge probability $p_{ij}$, allowing the model to explore new edges and resuscitate their sampling probabilities if necessary. We find that a $\delta$ as small as $0.01$ significantly helps in practice, and thus use this edge exploration scheme during training for our experiments.

**Wouldn't the model always prefer full attention?**   Note that the gradient $\partial \mathcal{L}/\partial p_{ij}$ can be positive, which suppresses the probability of edge $(i, j)$. At first, it may seem counter-intuitive why the model would ever limit itself to using fewer edges during training without any sparsity-based regularizations. One explanation is that masked attention provides an easy way to reduce attention scores under finite head dimensions. Under full attention, it is known that the representational space of attention score matrices is limited by the head dimension and softmax activation [5]. This limitation inevitably introduces unwanted noise in the attention scores especially when working with long sequences. In SBM-Transformer, however, the structural sparsity in masked attention introduces another dimension that induces a larger space of row-stochastic matrices (full attention is a special case of masked attention where $M_{ij} = 1$ for all $i, j$). Therefore, it is reasonable that the model may encourage sparsity to leverage the additional expressiveness assuming the loss landscape has local optima within the sparse attention regime. Our experiments on the LRA benchmark show that this is indeed the case, as our SBM-Transformer converges to an average attention sparsity of 20% to 30% while outperforming Transformer with full attention. We also show in the experiment that we can easily incorporate additional regularization that further encourages sparse attention masks.

### 4.3   SBM-Transformer is a Universal Approximator

Leveraging previous work on the theoretical expressiveness of sparse attention [45, 46], we show that SBM-Transformer with a small modification[1] retains the same level of expressibility as full attention. Specifically, we show that the low-rank structure of the underlying SBMs does not degrade the expressive power of Transformer, and that SBM-Transformer can universally approximate arbitrary functions with $\mathcal{O}(n)$ connections. For brevity, we provide a rough overview of the proof and defer further details to Appendix A.

**Theorem 1.** *Let $f \in \mathcal{F}$ be class of continuous sequence-to-sequence functions. $\mathcal{T}_{SBM}^{h,r,m}$ denote the class of SBM-Transformers with $h$ attention heads, $m$ head dimension, and $r$ dimensions in hidden layers. Then for any $\epsilon > 0$ and $1 \leq p < \infty$, there exists a function $g \in \mathcal{T}_{SBM}^{h,m,r}$ such that*

$$\int_{\mathbb{D}} \| f(\boldsymbol{X}) - \mathbb{E}[g(\boldsymbol{X})] \|_p^p d\boldsymbol{X} \leq \epsilon \tag{10}$$

---

[1]Here we consider a variant of SBM-Transformer where self-loops are added manually (i.e. $M_{ii} = 1$ for all $i$). While this is useful in theoretical analysis, we find that not having self-loops slightly helps in empirical performance and hence omit self-loops for the main experiments.

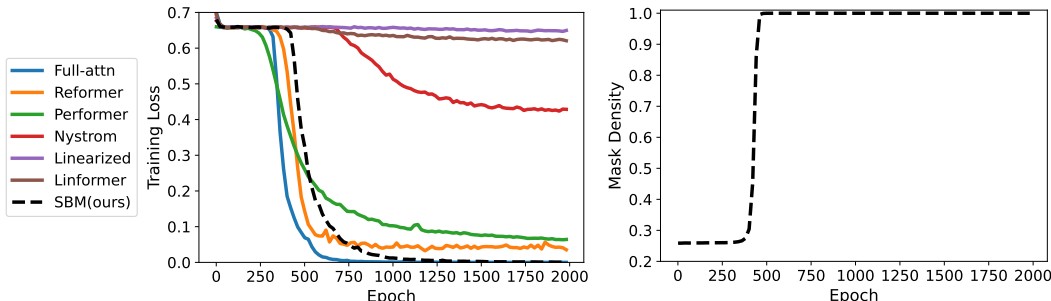

Figure 4: Loss (left) and mask density (right) of SBM-Transformer during training on the synthetic task. SBM-Transformer successfully converges to zero loss by tuning itself towards full attention.

According to the main theorem of Yun et al. (2020) [44], SBM-Transformer achieves universal approximability if 1) each node attends to itself, 2) the aggregation of all attention patterns contains a Hamiltonian path, and 3) there exists a path between all node pairs. While the first condition is trivially true due to our modification, the other two conditions require careful choice of three SBMs. Here we first parameterize one SBM to hard-assign tokens into $k$ equally-sized clusters, inducing a block-diagonal attention pattern. The other two SBMs are parameterized such that the two graphs together form a star graph with $k$ global relay tokens. Combining the three attention patterns lead to a parameterization of SBM-Transformer that satisfies all three conditions, hence proving the theorem.

## 5 Experiments

For empirical evaluations, we first use a synthetic task to show that our model is flexible enough to learn towards full attention when needed in contrast to previous works. We then experiment on Long Range Arena (LRA) [36], a benchmark widely used to assess the capacity of efficient Transformers in learning long-range contexts across different modalities. Lastly, we show results on the GLUE benchmark [39] to assess the performance of SBM-Transformer in a downstream NLP setting. All experiments were run on a remote GCP server equipped with 16 NVIDIA A100 Tensor Core GPUs.

### 5.1 Synthetic Task: Finding Repeated Tokens

**Dataset.** We formulate a token-level binary classification task as follows: each input sequence consists of $N$ integers, each of which is uniformly sampled from $\{1, 2, \ldots, N\}$. We use $N = 256$ in our setup. The prediction target is a sequence of equal length, where each token is labeled 1 if there exists a duplicate somewhere within the sequence, and 0 otherwise. Below is a simple example with $N = 8$ that illustrates the task. We measure the performance of models via binary cross-entropy loss.


Input: 1 4 3 7 3 2 3 1 ⇒ Target: 1 0 1 0 1 0 1 1


**Methods.** For this task, we compare SBM-Transformer with $k = 128$ clusters against various efficient Transformers: Linear Transformer [19], Linformer [40], Reformer [20], Performer [9], and Nyströmformer [43]. Across all methods, we use a single-layer and single-head architecture with 32 hidden dimensions. Note that due to this constrained setting, the sole head must perform full attention to compare each token to all the others in order to attain 100% accuracy. All models are trained for 2000 epochs where a new batch of sequences is sampled on-the-fly at each epoch. We use a batch size of 256 and learning rate of 1e-3.

**Results.** Figure 4 shows the training loss curves of each baseline method as well as SBM-Transformer. Full attention quickly converges to 100% accuracy, which is expected as it computes all possible pairwise interactions by default. Other models that apply low-rank or kernelized attention fail to achieve the same level of accuracy, due to limited expressibility under the constrained setting. Though SBM-Transformer converges more slowly compared to full-attention, it demonstrates the ability to drive itself towards full-attention, eventually attaining zero loss.

| Model | LISTOPS(2K) | TEXT(3K) | RETRIEVAL(4K) | IMAGE(1K) | PATHFINDER(1K) | Avg. |
|---|---|---|---|---|---|---|
| Full-attention [38] | 37.22 | 64.93 | 79.55 | 40.38 | 74.26 | 59.27 |
| Linearized [19] | 37.46 | 64.90 | **81.10** | 38.48 | 74.61 | 59.31 |
| Reformer [20] | 22.92 | 64.70 | 77.25 | **43.65** | 70.28 | 55.76 |
| Performer [9] | 18.25 | 65.00 | 79.01 | 39.80 | 70.79 | 54.57 |
| Linformer [40] | **38.44** | 56.28 | 78.09 | 39.53 | 67.62 | 55.99 |
| Nyströmformer [43] | 37.22 | 65.46 | 79.35 | 43.07 | 71.97 | 59.41 |
| SBM-Transformer (ours) | 37.45 (20.09%) | **65.79 (26.10%)** | 80.00 (29.46%) | 41.31 (20.49%) | **75.12 (18.56%)** | **59.93** |

Table 1: LRA benchmark results. The sequence lengths are shown next to each task. For SBM-Transformer, we report the average attention sparsity across all layers and heads during test time in parentheses. Bold and underlined results indicate best and 2nd best test accuracy for each task.

| $\lambda$ | LISTOPS(2K) | TEXT(3K) | RETRIEVAL(4K) | IMAGE(1K) | PATHFINDER(1K) | Avg. |
|---|---|---|---|---|---|---|
| 0 | 37.45 (20.09%) | **65.79 (26.10%)** | 80.00 (29.46%) | 41.31 (20.49%) | 75.12 (18.56%) | 59.93 |
| $10^{-4}$ | 37.76 (10.48%) | 65.48 (26.26%) | 79.93 (24.62%) | 41.35 (10.70%) | **75.46 (5.16%)** | **60.00** |
| $10^{-3}$ | **38.23 (10.46%)** | 65.18 (26.03%) | 80.00 (21.70%) | 41.17 (24.60%) | 74.49 (3.82%) | 59.81 |
| $10^{-2}$ | 38.20 (2.95%) | 65.59 (22.43%) | **80.44 (6.99%)** | **42.20 (3.95%)** | 72.79 (3.76%) | 59.84 |
| $10^{-1}$ | 37.76 (1.15%) | 64.48 (10.62%) | 79.46 (2.49%) | 41.35 (1.33%) | 73.79 (2.61%) | 59.37 |

Table 2: LRA results of SBM-Transformer with increasing sparsity regularization weight $\lambda$. Bold results indicate best accuracy for each task and percentage in parentheses indicate average attention density. Sparsity regularization helps in reducing computational cost with small drop in performance.

## 5.2 Long Range Arena (LRA)

To demonstrate that the flexible inductive bias of SBM-Transformer is effective for modeling long-range dependencies, we test SBM-Transformer against previous work on the LRA benchmark. We also test how the performance is affected with respect to applying a sparsity-based regularizer.

**Dataset.** LRA [36] consists of five different testbeds with varying modalities: LISTOPS [26] is a 10-way classification task to map a sequence of single-digit numbers and 4 different set operations, to its corresponding solution. TEXT [24] is a binary classification task where byte-level IMDB movie reviews must be classified into one of positive or negative sentiments. RETRIEVAL [30] is also a char-level binary classification task, where two sequences from ACL Anthology papers are given as input, and the model must predict whether there exists a citation link between them. IMAGE [21] is a 10-way classification task mapping flattened pixel-sequences from CIFAR-10 to its class. PATHFINDER [23] provides flattened pixel-sequences from an image and the model must decide whether two circles in the image are connected by a dashed line. For this benchmark, we use the PyTorch implementation of LRA provided by the authors of Nyströmformer [43] and adhere to the same train-test splits. Performance in all five tasks is measured using classification accuracy.

**Methods.** We compare SBM-Transformer against the same baselines as with the synthetic task above. For fair comparison, we set all Transformer models to use the default setting used in [43], which fixes 2 layers, 2 attention heads, and 64 embedding dimensions. For SBM-Transformer, we use $k = 128$ clusters. The output token representations are mean-pooled to obtain the sequence representation for all tasks. More details on the architecture setups can be found in Appendix C.

**Results.** Table 1 shows the test accuracies of each method. Our SBM-Transformer achieves the best overall performance, ranking first in two tasks, and second in one other. SBM-Transformer also outperforms full attention in all five tasks while computing 30% or less attention scores on average, which supports our claim that masked attention with partial attention score computations can be preferred over full attention depending on the task. With respect to the attention mask structure, we find that flexibility of SBM is indeed beneficial, as Reformer struggles in LISTOPS, most likely due to the inability of block-diagonal masks to model hierarchical contexts.

**Mask Density Regularization.** To test if the model can effectively learn under a constraint on the computational cost, we also test the model under a sparsity-based regularizer that discourages excessive use of query-key edges. We penalize each sampled edge by adding to the predictive

| Model | Relative FLOP Count | | | | | Relative Peak Memory Usage | | | | |
|---|---|---|---|---|---|---|---|---|---|---|
| | L(2K) | T(3K) | R(4K) | I(1K) | P(1K) | L(2K) | T(3K) | R(4K) | I(1K) | P(1K) |
| Full-attention [38] | 1.00 | 1.00 | 1.00 | 1.00 | 1.00 | 1.00 | 1.00 | 1.00 | 1.00 | 1.00 |
| Linearized [19] | 0.02 | 0.01 | 0.02 | 0.04 | 0.04 | 0.18 | 0.16 | 0.12 | 0.42 | 0.42 |
| Reformer [20] | 0.05 | 0.03 | 0.05 | 0.10 | 0.10 | 0.39 | 0.31 | 0.18 | 0.72 | 0.72 |
| Performer [9] | 0.18 | 0.12 | 0.18 | 0.36 | 0.36 | 0.76 | 0.70 | 0.60 | 0.96 | 0.96 |
| Linformer [40] | 0.33 | 0.22 | 0.33 | 0.66 | 0.66 | 0.26 | 0.22 | 0.14 | 0.34 | 0.34 |
| Nyströmformer [43] | 1.09 | 0.70 | 1.09 | 2.37 | 2.37 | 0.34 | 0.27 | 0.16 | 0.70 | 0.70 |
| SBM-Transformer (ours) | 0.07 | 0.23 | 0.08 | 0.27 | 0.29 | 0.19 | 1.01 | 0.19 | 0.39 | 0.48 |

Table 3: Per-example relative FLOP count and peak memory usage during LRA inference.

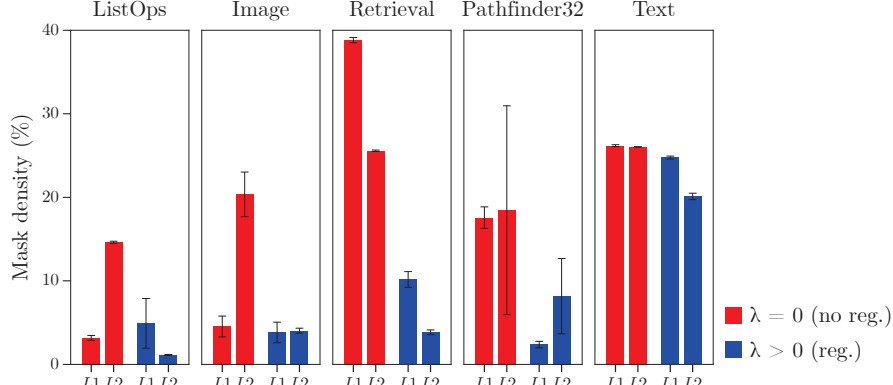

Figure 5: Average and standard deviation of density of masks sampled across the test set for each LRA task. The $x$-axis indicates the lower (L1) and upper (L2) layers and each bar represents the density averaged between the two attention heads in each layer.

loss a weighted regularization term $\lambda \mathcal{L}_s$, where $\mathcal{L}_s$ denotes the average mask density across all attention heads. Table 2 shows the performance of SBM-Transformer across varying regularization weights. Under strong regularization, the model surprisingly retains competitive performance while significantly reducing the average mask density. This indicates that similar local optima are shared across regimes with varying attention density in the loss landscape, and the regularization term is able to drive the model towards finding optimal attention scores with smaller density.

**Efficiency.** Furthermore, we compare computational costs during inference by measuring FLOP count and peak memory usage. For SBM-Transformer, we test the model trained under $\lambda = 10^{-1}$. Due to lack of support for sparse tensor operations in existing FLOP-counters, we measure FLOP counts by manually enumerating through each tensor operation. Table 3 shows that SBM-Transformer is comparably efficient across all tasks except for TEXT, where SBM-Transformer showed the largest average mask density. Note that while the cost of other baselines are fixed after initialization, the cost of SBM-Transformer is data-adaptive and can vary input-by-input. Further analysis and qualitative examples demonstrating the input-dependent attention mask densities can be found in Appendix C.

**Layerwise Diversity in Sparsity.** We also compare the densities of masks sampled at each layer of SBM-Transformer during test time to examine whether our model is capable of diversifying sparsity across layers for better performance. Recall that this allows models to gather information in different levels, as seen in pretrained BERT where lower layers focus on the overall content via dense attention while upper layers gather syntactic information with tree-like patterns [10]. For each of the five tasks, we pick two highest-performing models (one for unregularized and another for regularized) for measurement. Figure 5 shows the average layer-wise mask densities of unregularized and regularized SBM-Transformers across different tasks. We find that under no regularization, the two layers can differ by more than 10% in tasks such as LISTOPS and IMAGE. This may be due to the hierarchical and compositional structure of the two tasks. We also find that the variation is relatively low in TEXT with densities around 25%, indicating that the task requires broad attention overall. Lastly, the standard deviation is extremely large in upper layers for PATHFINDER, showing that it samples a wide variety of masks depending on the input.

## 5.3  General Language Understanding Evaluation (GLUE)

To check whether its strong performance demonstrated in LRA extends to the downstream NLP setting as well, we evaluate SBM-Transformer against baselines on the GLUE benchmark [39].

**Dataset.**  We consider four NLP tasks in GLUE [39]. SST-2 [35] consists of movie reviews the model must predict their positive or negative sentiments. For QQP [7], the task is to determine whether one question is a paraphrase of the other given a pair of questions. MNLI [42] consists of sentence pairs, each with a target label indicating whether the two sentences are connected through entailment, contradiction, or neither. QNLI [31] consists of sentence-question pairs and the task is to determine whether the sentence contains an answer to the question. Each task is formulated as sequence classification, and we measure performance by F1 score on the respective validation sets.

**Methods.**  Following previous work [43], we arrange a small variant of BERT [13] with 4 layers, 8 attention heads, and 512 embedding dimensions. We replace full attention with each attention module used in previous experiments. For SBM-Transformer, we use $k = 128$ clusters without sparsity regularization (i.e. $\lambda = 0$). Here, we find that adding local attention significantly boosts performance, and thus fix a sliding window of size 64 to SBM-Transformer. We first pretrain each model under the masked language modeling objective for 50 epochs on a corpus with text from English Wikipedia, BookCorpus [50], and RealNews [47]. We then finetune each pretrained model for 5 epochs on the GLUE training sets. More details on the architecture and training setup can be found in Appendix C.

**Results.**  Table 4 reports the F1 scores of each method on different NLP tasks. SBM-Transformer performs competitively against full attention overall, and outperforms all baselines in SST-2 and QQP. We also find that the fine-tuned SBM-Transformer models use 13.5% dense attention masks on average across all tasks, showing that the model can encode useful information from input sentences effectively under highly sparse attention.

| Model | SST-2 | QQP | MNLI | QNLI |
|---|---|---|---|---|
| Full-attention [38] | **89.8** | 84.7 | 84.0 | **85.0** |
| Reformer [20] | 89.3 | 84.4 | 83.9 | 84.0 |
| Performer [9] | 82.0 | 65.6 | 71.4 | 59.3 |
| Linformer [40] | 82.0 | 83.2 | 79.3 | 82.5 |
| Nyströmformer [43] | 89.7 | 83.2 | **84.1** | 84.9 |
| SBM-Transformer (ours) | **89.8** | **85.2** | 83.5 | 83.6 |

Table 4: GLUE benchmark results. Bold results indicate best accuracy for each task.

## 6  Conclusion

We propose SBM-Transformer, an efficient Transformer that can data-adaptively choose its attention sparsity between sparse and full attention without the need to explicitly compute the full attention score matrix. Theoretically, we show that our model enjoys the same expressibility as the original Transformer due to the flexibility of the latent SBM. Empirical experiments on LRA and GLUE show that our model performs competitively against previous state-of-the-art efficient Transformers.

Nonetheless, there are limitations due to sparse tensor operations being less optimized on GPU kernels. In the LRA experiments, we found that SBM-Transformer can result in longer runtimes compared to dense counterparts while its memory usage is much lower. While previous sparsity-based attention mechanisms with block-sparse attention are much more amenable for GPU computation [46, 8, 3], our work requires an architecture with better workload balancing and acceleration under unstructured sparsity, for which there is ongoing work [41, 49].

We still believe this work is valuable as it is the first approach to induce per-example attention sparsity, allowing the model to adjust its computational cost based on the input. The cost being dependent on the number of edges also allows practitioners to easily impose constraints based on the available computational resources. We hope to see more GPU-friendly tensor operations optimized for fine-grained sparsity in the future, at which point the value of this work will increase even further. As we propose a foundational replacement for the scaled dot-product attention module in the Transformer architecture, we do not expect any immediate negative societal impact due to this work.

## Acknowledgments and Disclosure of Funding

We would like to thank Kun Dong for the insightful comments. This work was supported by Institute of Information & communications Technology Planning & Evaluation (IITP) grant funded by the Korea government (MSIT) (No. 2022-0-00926, 2022-0-00959, 2021-0-02068, and 2019-0-00075).

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
