# OpenReview forum: "Transformers meet Stochastic Block Models: Attention with Data-Adaptive Sparsity and Cost"
_NeurIPS.cc/2022/Conference — NeurIPS 2022 Accept_

### Official Review · Reviewer_Nfpt · 2022-07-11

**Rating:** 6
**Confidence:** 4
**Soundness:** 2 fair
**Presentation:** 2 fair
**Contribution:** 2 fair

**Summary:**

The work under revision proposes SBM-Transformer. A novel variant of a transformer architecture adjusts a sparsity of its attention blocks dynamically, based on input sequence data. It is done so forward and backward evaluation cost is linear in the number of edges. Those are adaptively chosen by deploying a generative model for community detection, stochastic block model (SBM), combined with a fast sampling method, fastRG (Alg. 1, developed in [30]). Such an architecture is then flexible and capable of using a mixture of sparsities of attention blocks across the layers yet still capable of learning a full attention model if required. On average, however, it reduces computational complexity significantly, as documented in the well-chosen experimental section on the benchmark Long Range Arena (LRA) dataset.

The paper builds on previous theoretical expressivity results of sparse attention transformers and applies this result to proposed architecture in two Lemmas, resulting in Theorem 1 with proofs in the Appendix.

The paper concludes on the limitation of the method. Unfortunately, GPU architectures used to train large transformers are not efficient for sparse tensor operations. Proposed SMB-Transformer thus " ...often results in longer runtimes compared to dense counterparts even though the memory usage is lower ... ".

**Questions:**

To help accessibility of the article (see above) I suggest rewriting/extending the lines between 145 - 164. For instance why bipartite structure? Or what is 2-layer MLPd_h to d_h and why two layers (most likely to retain dimensionality but could you put some motivating lines of thoughts down?)

Are there any numerical or other limitations incurred by the use of SBM? In particular, this generative model assumes graph edges follow (mean parameterized) Poisson distribution. Besides one mentioned (line 182) regarding a degeneration of the mask matrix M, I'd encourage authors to elaborate more on this topic. For instance, is there any limitation in terms of learning "outliers" that is small clusters but highly connected clusters/tokens? (SMB is known to work well for the balanced size of clusters, but may struggle for imbalanced ones.)

Please find more suggestions in the previous block of comments.

**Limitations:**

A large part of the conclusions is dedicated to a major limitation - despite using fewer memory run-times on GPU clusters are often longer for SBM-Transformer than for dense models. In my opinion, this is unfortunate concerning the practical impact of this work because a majority of large transformer models GPT-3, PaLM, OPT175B, etc. have been trained in more and more accessible High-Performance Computing (HPC) centers, i.e. supercomputers, with massive GPU partitions.

Secondly all results hold "in expectation" (in a weak sense). Could authors elaborate more on the numerical stability/deterioration of the algorithm?

**Strengths And Weaknesses:**

Overall, this is an interesting application of sampling methods (an algorithm fastRG[30]) combined with gradient descent optimization with theoretical efficiency guarantees supported by experiments while not sacrificing the performance. To the reviewer's knowledge, this is a novel approach that I believe is of high interest to the ML community.

Short term, the practical impact/significance of the work is limited due to NOT being well suited for large GPU clusters - the nowadays platform of choice for large transformer models. Also mentioned in a nicely written Related Work, there are several methods in place, e.g., Reformer [18], using hashing similarity instead of SBM, similar in the idea, be it more constrained.

The paper could also be improved in terms of accessibility and readability, especially for a reader not familiar with SBM and generative modeling. For instance, it could help to elaborate in detail how a stochastic block model version from paper [2] (I believed that one is used), translates to SMB-transformer proposed. Especially in section 4.1.

Theorem 1: The idea of the proof presented in the main body of the paper is still technical and does not spark the intuition of why a “star graph” structure ensures expressibility after rephrasing the idea of the theorem from [38] on line 213. Moreover, the proof in Appendix relies on previous works, mainly by Yun et al.,(2020) [14], and Theorem 1 is proven by showing that SMB-Transformer meets the necessary assumptions of Yun et al.,(2020) [14]. Proof of Lemma 2 shows expressibility for p=3 sparsity patterns (expressibility is not guaranteed for p=2 as noted in line: 215) and thus proves it for higher p’s. However, it rather seems like a direct application of previously mentioned works on this special case of SMB-Transformer. Could you elaborate on what is a novelty here? Otherwise, I suggest being specific and making it an application of Theorem XY from [] and [] ...

line 285 and Table 2. Given the experimental std/errors (in brackets?) presented in the table, I am not sure we can conclude that “… applying a small sparsity regularizer helps in boosting overall sparsity as well as performance …”.  I would claim it is rather insignificant looking at average results in the last column! I agree, however, the computation saving effect of regularization vs. no reg. in Figure 5 still holds. So I suggest considering rephrasing or omitting the claim above perhaps.

Minor comments:
typo: typo Table 1 “the the”

---

> ### Author Response · Authors · 2022-08-02
> **Official Response to Reviewer Nfpt (1/2)**
>
> ---
>
> > The practical impact/significance of the work is limited due to NOT being well suited for large GPU clusters - the nowadays platform of choice for large transformer models. A large part of the conclusions is dedicated to a major limitation - despite using fewer memory run-times on GPU clusters are often longer for SBM-Transformer than for dense models.
>
> &rarr; As mentioned by the reviewer, GPU kernels are less optimized for operations on tensors with unstructured sparsity. However, recent works have proposed ways to achieve faster computation via efficient mapping onto Tensor Cores and load balancing across threads [A,B]. We hope to see such methods incorporated into deep learning libraries, at which point the architecture-related bottleneck will become insignificant.
>
> [A] Ziheng Wang, "SparseRT: Accelerating Unstructured Sparsity on GPUs for Deep Learning Inference" ACM2020
>
> [B] Zhu et al., "Taming Unstructured Sparsity on GPUs via Latency-Aware Optimization" ACM/IEEE2020
>
> ---
>
> > The paper could also be improved in terms of accessibility and readability, especially for a reader not familiar with SBM and generative modeling. For instance why bipartite structure? Or what is 2-layer $MLP_{d_h \to d_h}$ and why two layers (most likely to retain dimensionality but could you put some motivating lines of thoughts down?)
>
> &rarr; We will revise the text lines 145 to 164 for better accessibility on how the SBM generates graphs as well as connections to the attention mechanism. The answers to the two specific questions are as below, and we will add this information in the revision.
> - The bipartite structure is assumed due to the natural formulation of the attention mechanism: each query attending to a key can be represented as a directed edge from the query node to the key node. Note that the bipartite structure is useful also for cross-attention where there can be different numbers of queries and keys. In such cases, a general undirected graph does not suffice, while the directed bipartite graph structure fits right in.
> - The 2-layer MLP allows the model to learn a richer space of node embeddings that are detached from the token-embedding space. Through ablation studies with variants using a shallower 1-layer MLP or no MLP for token-to-node projection, we found that the 2-layer MLP is necessary for better performance. Note that having the additional MLPs only slightly increases the number of parameters as well as FLOP counts.
>
> |  | ListOps | Text | Retrieval | Image | Pathfinder |
> | ---: | :---: | :---: | :---: | :---: | :---: |
> | 2-layer MLP  | 37.50 | 65.79 | 80.00 | 41.31 | 75.12 |
> | 1-layer MLP  | 18.50 | 64.72 | 73.37 | 39.95 | 67.51 |
> | 0-layer MLP  | 17.95 | 64.41 | 73.12 | 40.32 | 68.21 |
>
> ---
>
> > Theorem 1: The idea of the proof presented in the main body of the paper is still technical and does not spark the intuition of why a “star graph” structure ensures expressibility after rephrasing the idea of the theorem from [38] in line 213.
>
> &rarr; According to Lemma 3 of [C], having one global token connected to all other tokens is required for self-attention to encode the entire context of the input sequence. In terms of the attention patterns, the aggregation of all attention patterns must contain the star graph, where a global token $x_0$ is connected to all tokens $\mathcal{N}(x_0) = \{x_0,x_1,\dots,x_n\}$, and any other token $x_i$ is connected to the global token and itself (i.e. $\mathcal{N}(x_i) = \{x_0, x_i\}$). In summary, the proof of Theorem 1 in the appendix shows that despite the low-rank structure of the underlying SBM, there exists a parameterization of the SBM-Transformer such that the aggregated attention pattern contains the star graph. The approach we use to prove the theorem is to separate the star graph into three components, each of which is representable by a single SBM. We will improve the text following Theorem 1 to clarify this intuition.
>
> [C] Zaheer et al., "Big Bird: Transformers for Longer Sequences" NeurIPS2020
>
> ---
>
> > The proof of Theorem 1 seems like a direct application of previously mentioned works on this special case of SMB-Transformer. Could you elaborate on what is a novelty here?
>
> &rarr; As noted by the reviewer, our Theorem is derived from previous theoretical work on the expressibility of sparse attention. However, our contribution is that the low-rank structure of the underlying SBM does not deteriorate the expressive power of the model. We believe that this is still significant to practitioners, as it provides a quick sanity check that the model retains the same theoretical power as full-attention. We will clarify this significance in the revision.

---

> > ### Author Response · Authors · 2022-08-02
> > **Official Response to Reviewer Nfpt (2/2)**
> >
> > ---
> >
> > > Line 285 and Table 2. Given the experimental std/errors (in brackets?) presented in the table, I am not sure we can conclude that “… applying a small sparsity regularizer helps in boosting overall sparsity as well as performance …”. I would claim it is rather insignificant looking at average results in the last column!
> >
> > &rarr; We agree that the performance gain is rather insignificant, but the gain in computational cost while showing comparable performance is still significant as mentioned by the reviewer. We will revise the sentence to avoid misleading the readers.
> >
> > ---
> >
> > > Minor comments: typo: typo Table 1 “the the”
> >
> > &rarr; Thank you for the comment, we will correct the typo in the revision.
> >
> > ---
> >
> > > Are there any numerical or other limitations incurred by the use of SBM? Is there any limitation in terms of learning "outliers" that is small clusters but highly connected clusters/tokens? (SMB is known to work well for the balanced size of clusters, but may struggle for imbalanced ones.)
> >
> > &rarr; As mentioned by the reviewer, recovering ground-truth communities is indeed theoretically hard when the node-memberships are unbalanced [D]. However, this applies when the problem is to infer latent parameters of the SBM given the graph, while in our framework, the process is in reverse where the parameters are first produced based on the input tokens, and the graph is sampled afterward. Thus the problem with unbalanced clusters is not detrimental in our case, and the sampled graph can be flexibly consisted of large as well as small clusters. We will elaborate more on this topic in the revision when discussing the SBM parametrization.
> >
> > [D] Emmanuel Abbe, "Community Detection and Stochastic Block Models:
> > Recent Developments" JMLR2018
> >
> > ---
> >
> > > All results hold "in expectation". Could authors elaborate more on the numerical stability/deterioration of the algorithm?
> >
> > &rarr; Even though SBM-Transformer is stochastic even during inference due to the graph sampling step, we found that the predictions resulting from the model are stable despite this randomness (leading to relatively low standard deviations in LRA accuracy compared to other baselines as shown in our main results).
> >
> > To further support this claim, we measure the prediction stability as follows: For each LRA task, we measure the percentage of test examples that the model predicts correctly the first time, and also predicts correctly during a second inference. We find that the stability exceeds 95% across all tasks, showing that SBM-Transformer makes correct predictions consistently overall, despite the stochasticity in its attention mask sampling step.
> >
> > |  | ListOps | Text | Retrieval | Image | Pathfinder |
> > | ---: | :---: | :---: | :---: | :---: | :---: |
> > | Prediction Stability  | 98.34% | 99.10% | 99.68% | 95.55% | 96.65% |

---

### Official Review · Reviewer_sh4F · 2022-07-11

**Rating:** 6
**Confidence:** 4
**Soundness:** 2 fair
**Presentation:** 3 good
**Contribution:** 2 fair

**Summary:**

This paper proposes a Transformer variant where the self-attention matrices are masked using random adjacency matrices generated using a stochastic block model (SBM). The goal here is to reduce the computational cost of producing each N x N self-attention matrix. In the forward pass, the proposed method computes SBM parameters from the query and key embeddings using an MLP and samples a self-attention mask using a fast random graph sampling method. In the backward pass, gradients are passed through this discrete sampling step using a straight-through estimator. The authors show that the proposed SBM-Transformer is a universal approximator in the setting considered by Yun et al. (2020) and Zaheer et al. (2020). Empirically, the authors demonstrate that the SBM-Transformer (1) is able to emulate a standard Transformer with dense attention on a synthetic task and (2) outperforms several efficient Transformer variants on average on the Long Range Arena benchmark.

**Questions:**

- As discussed in the previous section, does the SBM-Transformer push out the accuracy-cost Pareto frontier vs. existing efficient Transformer architectures?
- Nondeterministic prediction at inference time is typically considered to be undesirable by practitioners. However, the setup as described requires random sampling under the SBM even during inference. Can this sampling step by replaced with a deterministic operation?


**Limitations:**

The authors highlight the limitation of their work with regard to hardware support for unstructured sparsity in Section 6.

**Strengths And Weaknesses:**

### Originality

- As noted in the paper, the use of sparse attention masks to reduce the computational cost of the self-attention operation has previously been explored in several existing papers (e.g., axial attention (Ho et al. 2019), BigBird (Zaheer et al., 2020)). The main methodological novelty here is the use of a stochastic block model to generate a random attention mask in each forward pass. To my knowledge, this approach has not been studied in prior work.

### Quality

- **Theoretical results.** The paper establishes a universal approximation property for the proposed transformer variant (Theorem 1). This result is helpful as a sanity check that the proposed method is a reasonable approach. The overall technical contribution here is relatively minor since the proof of the result is primarily an application of existing machinery from Yun et al. (2020) and Zaheer et al. (2020). A minor weakness of the analysis is that it makes use of a modified version of the SBM-Transformer architecture relative to that used in the empirical evaluation -- in particular, the analysis requires that M_ii = 1, unlike in the version used to obtain the results in Table 1.
- **Empirical evaluation.** A strength of the paper is the result that the proposed SBM-Transformer architecture outperforms the baseline efficient Transformer architectures on average over the LRA benchmark tasks (Table 1). However, a significant weakness of this comparison is that it does not appropriately account for the accuracy-cost tradeoff corresponding to each of the evaluated methods. For instance, it is not obvious from the paper whether the SBM-Transformer outperforms the given baselines when controlling for a measure of computational cost such as the per-example FLOP count during inference. Such controls are important since the SBM-Transformer introduces additional operators (such as a 2-layer MLP used for computing the SBM parameters), the costs of which will have to be offset by the increased sparsity of the self-attention matrix.

### Clarity

- The presentation was generally clear and easy to follow.
- The specification of the masked attention computation in Section 3.2 is problematic since simply masking the attention logits to 0 is not equivalent to zeroing-out the attention scores after the softmax operator.

### Significance

- As noted in Section 6, the unstructured sparsity of the resulting attention maps in the SBM-Transformer formulation is not amenable to fast matrix multiplication on current GPU hardware. Thus, as a practical matter, the proposed method requires longer wall clock times for inference, which consequently limits the significance of the work for practitioners. The authors argue that the method (1) "enables multi-head attention with long sequences", and that (2) practitioners can tune the computational cost of inference by imposing constraints on the number of edges to sample. I don't find these arguments to be convincing as is -- in particular, claim (1) requires additional justification in light of (a) existing efficient transformer architectures that incur subquadratic computation, and (b) techniques such as memory-efficient attention [1] that require subquadratic memory. As for claim (2), the merit of this additional flexibility (e.g., in terms of accuracy vs. cost) should be evaluated empirically.

[1] Rabe & Staats, 2021. Self-attention Does Not Need O(n^2) Memory.

---

> ### Author Response · Authors · 2022-08-02
> **Official Response to Reviewer sh4F (1/3)**
>
> ---
>
> > A minor weakness of the analysis is that it makes use of a modified version of the SBM-Transformer architecture relative to that used in the empirical evaluation
>
> &rarr; In the original supplementary material, we have presented LRA results of the SBM-Transformer variant with self-loops added. We find that the practical impact of adding self-loops is rather minor, while the one without self-loops performs slightly better. We will clarify this distinction in the revision.
>
> ---
>
> > A significant weakness of this comparison is that it does not appropriately account for the accuracy-cost tradeoff corresponding to each of the evaluated methods. Does the SBM-Transformer push out the accuracy-cost Pareto frontier vs. existing efficient Transformer architectures?
>
> &rarr; To illustrate the overall cost of SBM-Transformer, we report FLOP counts (per example and attention head), given the hyperparameter setting used in our LRA experiments. For SBM-Transformer, we measure the cost of models trained with $\lambda=10^{-1}$ with task performances reported in Table 2. Due to difficulty in applying FLOP-counting toolkits to custom modules, FLOP counts are instead measured by enumerating through each tensor operation (addition and multiplication) while excluding tensor manipulations such as ``reshape`` and ``stack``. We also report the relative peak memory use, which we measured empirically during LRA test time.
>
> | Relative FLOP count | ListOps | Text | Retrieval | Image | Pathfinder |
> | ---: | :---: | :---: | :---: | :---: | :---: |
> | Full Attention  | 1.000 | 1.000 | 1.000 | 1.000 | 1.000 |
> | Linearized | 0.021 | 0.014 | 0.021 | 0.042 | 0.042 |
> | Reformer | 0.049 | 0.033 | 0.049 | 0.098 | 0.098 |
> | Performer  | 0.182 | 0.121 | 0.182 | 0.363 | 0.363 |
> | Nyströmformer  | 1.086 | 0.701 | 1.086 | 2.374 | 2.374 |
> | Linformer   | 0.328 | 0.219 | 0.328 | 0.656 | 0.656 |
> | SBM-Transformer (ours)  | 0.068 | 0.225 | 0.079 | 0.273 | 0.290 |
>
> | Relative Peak Memory | ListOps | Text | Retrieval | Image | Pathfinder |
> | ---: | :---: | :---: | :---: | :---: | :---: |
> | Full Attention  | 1.00 | 1.00 | 1.00 | 1.00 | 1.00 |
> | Linearized  | 0.18 | 0.16 | 0.12 | 0.42 | 0.42 |
> | Reformer  | 0.39 | 0.31 | 0.18 | 0.72 | 0.72 |
> | Performer  | 0.76 | 0.70 | 0.60 | 0.96 | 0.96 |
> | Nyströmformer  | 0.34 | 0.27 | 0.16 | 0.70 | 0.70 |
> | Linformer   | 0.26 | 0.22 | 0.14 | 0.34 | 0.34 |
> | SBM-Transformer (ours)  | 0.19 | 1.01 | 0.19 | 0.39 | 0.48 |
>
> According to the results above and in the main paper, we find that SBM-Transformer does not push out the Pareto-frontier by a significant amount. However, we believe its strength lies in allowing the model to autonomously choose from the set of all Pareto efficient solutions based on the task. While all other methods are restricted to a search space with fixed cost once the model size is fixed, SBM-Transformer allows more flexible optimization due to the attention mask density being dependent upon the input.
>
> ---
>
> > The specification of the masked attention computation in Section 3.2 is problematic since simply masking the attention logits to 0 is not equivalent to zeroing-out the attention scores after the softmax operator.
>
> &rarr; The $\sigma_\mathbf{M}(\cdot)$ operation shown in Equation 4 denotes a masked softmax operator, and is computed for the $(i,j)$-th entry only if $\mathbf{M}\_{ij} = 1$. All other entries remain as zero. Note the normalization term also only sums up entries with mask value 1. In all, Equations 3 and 4 are together equivalent to a dense operation that fills negative infinity in all $(i,j)$-th entries with $\mathbf{M}\_{ij} = 0$, and applying the standard softmax. We will add clarifications to avoid confusion.

---

> > ### Author Response · Authors · 2022-08-02
> > **Official Response to Reviewer sh4F (2/3)**
> >
> > ---
> >
> > >  As a practical matter, the proposed method requires longer wall clock times for inference, which consequently limits the significance of the work for practitioners.
> >
> > &rarr; As mentioned by the reviewer, we find that the empirical runtime is \~x4 larger than that of full-attention in LRA. The main bottlenecks in our current PyTorch implementation are two-fold: the graph sampling procedure with ``fastRG`` (which contributes \~58% runtime) and 2) the following sparse tensor operations (\~42% runtime). There is also the architecture-related bottleneck due to the lack of GPU-friendly tensor operations under unstructured sparsity. Nonetheless, we believe such bottlenecks will soon become negligible in the near future, and provide further explanations for each issue below:
> >
> > 1. *Batched node sampling with PyTorch*: At its current state, PyTorch only supports sampling the same number of samples per distribution (https://github.com/pytorch/pytorch/issues/42407). Thus to parallelize node sampling in ``fastRG``, our current implementation first finds the maximum number of samples needed per cluster, and then samples the same number of nodes from all node-distributions. We then apply a mask to remove the excess samples. Naturally, this incurs unnecessary costs that can be large especially when there is a big gap among the number of samples to take from each distribution. The functionality of sampling a different number of samples from different distributions efficiently is implemented in Numpy (https://numpy.org/doc/stable/reference/random/generated/numpy.random.Generator.multinomial.html), and we expect a similar implementation to be available soon in Pytorch as well.
> > 2. *Sparse Tensor Operations in PyTorch*: our graph sampling step with ``fastRG`` returns an edgelist, which naturally fits in the coordinate-format (COO) sparse tensor in the ``torch.sparse`` module. However, the ``sparse.sampled_addmm`` operator that sparsely computes the masked dot product only supports compressed sparse row-format (CSR), while ``sparse.softmax`` only supports COO. This causes additional cost of converting the format of the sparse tensor back and forth. Considering that the ``torch.sparse`` module is yet in its beta state (https://pytorch.org/docs/stable/sparse.html), we expect operations to soon be applicable in both CSR and COO format, at which point this conversion will become unnecessary.
> > 3. *GPU Kernels for Unstructured Sparsity*: As mentioned in the conclusion of our submission, GPU kernels are less optimized for operations on tensors with unstructured sparsity. However, recent works have proposed ways to achieve faster computation via efficient mapping onto Tensor Cores and load balancing across threads [A,B]. We hope to see such methods incorporated into deep learning libraries, at which point the architecture-related bottleneck will become insignificant.
> >
> > [A] Ziheng Wang, "SparseRT: Accelerating Unstructured Sparsity on GPUs for Deep Learning Inference" ACM2020
> >
> > [B] Zhu et al., "Taming Unstructured Sparsity on GPUs via Latency-Aware Optimization" ACM/IEEE2020
> >
> > ---
> >
> > > The authors argue that the method (1) "enables multi-head attention with long sequences", and that (2) practitioners can tune the computational cost of inference by imposing constraints on the number of edges to sample. I don't find these arguments to be convincing as is.
> >
> > &rarr; We agree that the arguments may mislead the main strengths of our work. Our original intention was to state that SBM-Transformer can 1) flexibly learn long-range dependencies with data-adaptive sparsity and cost and also 2) allow users to easily impose a constraint on computation cost by controlling the number of edges to sample. We will correct the statement in the revision.

---

> > > ### Author Response · Authors · 2022-08-02
> > > **Official Response to Reviewer sh4F (3/3)**
> > >
> > > ---
> > >
> > > > Nondeterministic prediction at inference time is typically considered to be undesirable by practitioners. Can the sampling step be replaced with a deterministic operation during inference?
> > >
> > > &rarr; One simple deterministic approach could be to use a $k$-nearest neighbor approach and connect each query to $k$ keys with the highest edge probabilities. Leveraging the parameterization of edge probabilities as $p(i,j) = \hat{\mathbf{Q}}_i\hat{\mathbf{S}}\hat{\mathbf{K}}_j^T$ (line 156), finding the $k$-nearest neighbors can be done in cost linear to the sequence length. However, this approach not only requires a fixed number of neighbors to use as a hyperparameter, but also imposes a strong constraint on the graph structure (each query has $k$ edges) that can potentially deteriorate performance.
> > >
> > > To show that nondeterministic prediction does not largely impact the overall performance of the SBM-Transformer, we report the prediction stability as follows: For each LRA task, we measure the percentage of test examples that the model predicts correctly the first time, and also predicts correctly during the second inference. We find that the stability exceeds 95% across all tasks, showing that SBM-Transformer makes correct predictions consistently overall despite the randomness in its attention mask sampling step. This is also supported by the low standard deviations in accuracy reported in Tables 4 and 5 of the Appendix.
> > >
> > > |  | ListOps | Text | Retrieval | Image | Pathfinder |
> > > | ---: | :---: | :---: | :---: | :---: | :---: |
> > > | Prediction Stability  | 98.34% | 99.10% | 99.68% | 95.55% | 96.65% |

---

> > ### Comment · Reviewer_sh4F · 2022-08-03
> > **Response to authors**
> >
> > (Responding to the parent comment to avoid deeper nesting)
> >
> > Thanks for the detailed response and for providing additional measurements. Upon further consideration, I've decided to raise my score from 4 to 6. I think that the conceptual idea of achieving adaptive per-example sparsity using a cheaper low-rank computation (essentially a kind of "cascade" technique applied to the attention matrix) is interesting, and I didn't give enough credit to this aspect of the work in the initial review. Additionally, I recognize that hardware and low-level kernel support for sparse operations is a challenge for this sort of work at present, and I appreciate the authors' response on this point.

---

> > > ### Author Response · Authors · 2022-08-05
> > > **Thank You to Reviewer sh4F**
> > >
> > > Thank you for the response. We are glad that you found our idea interesting, and we will post a revision addressing your comments soon. Please let us know in case you need further clarifications afterwards. Thank you!

---

### Official Review · Reviewer_ukVv · 2022-07-12

**Rating:** 6
**Confidence:** 4
**Soundness:** 3 good
**Presentation:** 4 excellent
**Contribution:** 2 fair

**Summary:**

In this paper, the authors propose to adaptively learn the sparsity of attention module through modeling the mask matrix as a bipartite graph generated by SBM.

Specifically, they discuss how they can apply the new attention mechanism to both forward and backward propagation.
- They use Eqn. (5), (6), (7), and (8) to model the mask matrix in forward propagation, which introduce a trainable cluster embedding matrix $C$ and two MLPs to decide the node membership of tokens in $Q, K$.
- They "incorporate a Straight-Through Estimator (STE) to pass the gradient beyond the discrete sampling", which works as if $M \odot \mathbb E M$ is used in forward propagation.
- They also show the new attention mechanism is a universal approximator, and evaluate their proposed methods on Long-range-arena (LRA) benchmarks.

**Questions:**

Could the authors show the extra time and space cost of adding the SBM mechanism? How is that compared to another efficient attention mechanism?

**Strengths And Weaknesses:**

### Originality

The idea to model the sparsity of attention by SBM is new, though formally similar to the classical idea in pruning that directly model the whole mask matrix.

### Quality

Pros:

- The author propose a new data-adaptive sparse attention mechanism.


Cons:
- The experiments on LRA is not that convincing nowadays given the recent paper "Efficiently Modeling Long Sequences with Structured State Spaces" (ICLR 2022), which shows current Transformers are way worse than their state-space-based method. The effectiveness of the experimental results would be in doubt whether in another setting/benchmarks which Transformers are more good at, the conclusion would still hold.
- The actual runtime of the new method is not reported and compared to exisitng methods.


### Clarity

The clarity of this paper is awesome. There is a corner concern: to my knowledge, "SBM" should be short for "stochastic block model" rather than stochastic blockmodel.

Also, the statement of Theorem 1 is not very informative. It would be better if the content in Appendix A that Eqn. (10) can hold with $O(n)$ connections can be moved to the main paper.

### Significance

The problem studied is important: how to sparsify and therefore accelerate attention. However, the GPU-unfriendly sparse matrix multiplication makes the significance limited.

---

> ### Author Response · Authors · 2022-08-02
> **Official Response to Reviewer ukVv (1/2)**
>
> ---
>
> > The effectiveness of experimental results would be in doubt whether in another setting/benchmarks which Transformers are better than state-space-based methods, the conclusion would still hold.
>
> &rarr; To further support the competitive performance of our method, we also evaluate the performance of all methods on NLP downstream tasks on the GLUE benchmark [A], following the same setting used in the Nyströmformer paper [B]. We first pretrain a BERT-small (4 layers and 8 heads) model using each different attention module for 50 epochs on a dataset consisting of texts from Wikipedia, BookCorpus, and RealNews. We then finetune each model on four different tasks in GLUE for 5 epochs, and report the F1 scores on the respective validation sets below. Note that we were not able to measure the performance of linearized attention as it failed to converge properly during pretraining.
>
> The table below shows the results on the GLUE benchmark. We find that SBM-Transformer performs competitively across all efficient Transformer variants, even surpassing the performance of full attention on one task. Considering that the average attention mask density of the SBM-Transformer is 13.5% across all tasks, its performance shows that the underlying SBM structure is useful in encoding contextual mappings under low cost. We will add these results in the experiment section.
>
> |  | SST-2 | QNLI | QQP | MNLI |
> | ---: | :---: | :---: | :---: | :---: |
> | Full Attention  | **89.8** | **85.0** | 84.7 | 84.0 |
> | Linearized | N/A | N/A | N/A | N/A |
> | Reformer   | 89.3 | 84.0 | 84.4 | 83.9 |
> | Performer  | 82.0 | 59.3 | 65.6 | 71.4 |
> | Nyströmformer  | 89.7 | 84.9 | 83.2 | **84.1** |
> | Linformer   | 82.0 | 82.5 | 83.2 | 79.3 |
> | SBM-Transformer (ours)  | **89.8** | 83.6 | **85.2** | 83.5 |
>
> [A] Wang et al., "GLUE: A Multi-Task Benchmark and Analysis Platform for Natural Language Understanding" ICLR 2019
>
> [B] Xiong et al., "Nyströmformer: A Nyström-Based Algorithm for Approximating Self-Attention" AAAI 2021
>
> ---
>
> > The actual runtime of the new method is not reported and compared to existing methods. Could the authors show the extra time and space cost of adding the SBM mechanism? How is that compared to another efficient attention mechanism?
>
> &rarr; To illustrate the overall cost of SBM-Transformer, we report FLOP counts (per example and attention head), given the hyperparameter setting used in our LRA experiments. For SBM-Transformer, we measure the cost of models trained with $\lambda=10^{-1}$ with task performances reported in Table 2. Due to difficulty in applying FLOP-counting toolkits to custom modules, FLOP counts are instead measured by enumerating through each tensor operation (addition and multiplication) while excluding tensor manipulations such as ``reshape`` and ``stack``. We also report the relative peak memory use, which we measured empirically during LRA test time.
>
> | Relative FLOP count | ListOps | Text | Retrieval | Image | Pathfinder |
> | ---: | :---: | :---: | :---: | :---: | :---: |
> | Full Attention  | 1.000 | 1.000 | 1.000 | 1.000 | 1.000 |
> | Linearized | 0.021 | 0.014 | 0.021 | 0.042 | 0.042 |
> | Reformer | 0.049 | 0.033 | 0.049 | 0.098 | 0.098 |
> | Performer  | 0.182 | 0.121 | 0.182 | 0.363 | 0.363 |
> | Nyströmformer  | 1.086 | 0.701 | 1.086 | 2.374 | 2.374 |
> | Linformer   | 0.328 | 0.219 | 0.328 | 0.656 | 0.656 |
> | SBM-Transformer (ours)  | 0.068 | 0.225 | 0.079 | 0.273 | 0.290 |
>
> | Relative Peak Memory | ListOps | Text | Retrieval | Image | Pathfinder |
> | ---: | :---: | :---: | :---: | :---: | :---: |
> | Full Attention  | 1.00 | 1.00 | 1.00 | 1.00 | 1.00 |
> | Linearized  | 0.18 | 0.16 | 0.12 | 0.42 | 0.42 |
> | Reformer  | 0.39 | 0.31 | 0.18 | 0.72 | 0.72 |
> | Performer  | 0.76 | 0.70 | 0.60 | 0.96 | 0.96 |
> | Nyströmformer  | 0.34 | 0.27 | 0.16 | 0.70 | 0.70 |
> | Linformer   | 0.26 | 0.22 | 0.14 | 0.34 | 0.34 |
> | SBM-Transformer (ours)  | 0.19 | 1.01 | 0.19 | 0.39 | 0.48 |
>
> Results show that SBM-Transformer uses cost comparable to those of other baselines. Note that measurements for SBM-Transformer can change per input as the attention sparsity is data-adaptive. Examples demonstrating this unique feature of SBM-Transformer can be found in Figures 2 and 3 of the Appendix.

---

> > ### Author Response · Authors · 2022-08-02
> > **Official Response to Reviewer ukVv (2/2)**
> >
> > Unfortunately, we find that the empirical runtime is \~x4 larger than that of full-attention in LRA. The main bottlenecks in our current PyTorch implementation are two-fold: the graph sampling procedure with ``fastRG`` (which takes \~58% runtime) and 2) the sparsely sampled matrix multiplication followed by sparse softmax operation (\~42% of runtime). There is also the architecture-related bottleneck due to the lack of GPU-friendly tensor operations under unstructured sparsity. Nonetheless, we believe such bottlenecks will soon become negligible in the near future, and provide further explanations for each issue below:
> >
> > 1. *Batched node sampling with PyTorch*: At its current state, PyTorch only supports sampling the same number of samples per distribution (https://github.com/pytorch/pytorch/issues/42407). Thus to parallelize node sampling in ``fastRG``, our current implementation first finds the maximum number of samples needed per cluster, and then samples the same number of nodes from all node-distributions. We then apply a mask to remove the excess samples. Naturally, this incurs unnecessary costs that can be large especially when there is a big gap among the number of samples to take from each distribution. The functionality of sampling a different number of samples from different distributions efficiently is implemented in Numpy (https://numpy.org/doc/stable/reference/random/generated/numpy.random.Generator.multinomial.html), and we expect a similar implementation to be available soon in Pytorch as well.
> > 2. *Sparse Tensor Operations in PyTorch*: our graph sampling step with ``fastRG`` returns an edgelist, which naturally fits in the coordinate-format (COO) sparse tensor in the ``torch.sparse`` module. However, the ``sparse.sampled_addmm`` operator that sparsely computes the masked dot product only supports compressed sparse row-format (CSR), while ``sparse.softmax`` only supports COO. This causes additional cost of converting the format of the sparse tensor back and forth. Considering that the ``torch.sparse`` module is yet in its beta state (https://pytorch.org/docs/stable/sparse.html), we expect operations to soon be applicable in both CSR and COO format, at which point this conversion will become unnecessary.
> > 3. *GPU Kernels for Unstructured Sparsity*: As mentioned in the conclusion of our submission, GPU kernels are less optimized for operations on tensors with unstructured sparsity. However, recent works have proposed ways to achieve faster computation via efficient mapping onto Tensor Cores and load balancing across threads [C,D]. We hope to see such methods incorporated into deep learning libraries, at which point the architecture-related bottleneck will become insignificant.
> >
> > [C] Ziheng Wang, "SparseRT: Accelerating Unstructured Sparsty on GPUs for Deep Learning Inference" ACM2020
> >
> > [D] Zhu et al., "Taming Unstructured Sparsity on GPUs via Latency-Aware Optimization" ACM/IEEE2020
> >
> > ---
> >
> > > "SBM" should be short for "stochastic block model" rather than stochastic blockmodel.
> >
> > &rarr; We will make changes in the revision to avoid confusion.
> >
> > ---
> >
> > > The statement of Theorem 1 is not very informative.
> >
> > &rarr; We appreciate the comment. We agree that adding the content of the proof will provide further insight on how SBM-Transformer preserves universal expressibility with $\mathcal{O}(n)$ connections. We will move the analysis to the main paper for better clarity.
> >
> > ---
> >
> > > However, the GPU-unfriendly sparse matrix multiplication makes the significance limited.
> >
> > &rarr; As mentioned above, recent works have proposed methods for accelerating GPU-computation with unstructured sparsity (e.g. SparseRT [E]). We believe this line of work has just started to show significant speedup under unstructured sparsity, and the flexibility and performance of the SBM-Transformer will soon outweigh its computational cost in the near future.
> >
> > [E] Ziheng Wang. "SparseRT: Accelerating Unstructured Sparsity on GPUs for Deep Learning Inference" ACM2020

---

> ### Comment · Reviewer_ukVv · 2022-08-08
> **Response**
>
> I have read the response and the other reviews I have a better understanding of the work. Considering the limited practical performance on GPU I'm happy with my original rating. I appreciate the organized response and the new experimental results added by the authors.

---

> > ### Author Response · Authors · 2022-08-08
> > **Thank You to Reviewer ukVv**
> >
> > Thank you for the response. Just as a friendly reminder, we have also uploaded a revision of our manuscript to address your comments and concerns. Please feel free to take a look and let us know in case of any further questions. Thank you!

---

### Author Response · Authors · 2022-08-02
**Common Response to All Reviewers**

We sincerely thank all reviewers for the constructive reviews and suggestions. Overall, we are very encouraged by the reviewers' positive feedback on the novelty and clarity of our work. With regards to specific questions on the practical significance and needed clarifications within our work, our responses can be found in the respective comments below.

---

### Author Response · Authors · 2022-08-07
**Uploaded First Revision**

To address comments and concerns from the reviewers, we have made the following updates on our main manuscript:
- Overall: Changed "Stochastic Blockmodel" to "Stochastic Block Model" (for reviewer ukVv)
- Lines 122-125: Added clarifications on the masked-softmax operator in Equation 4 (for reviewer sh4F)
- Lines 205-221: Added clarifications on the significance of Theorem 1 and the brief sketch of the proof (for reviewers ukVv, Nfpt)
- Lines 205-210: Mentioned that we consider a variant of SBM-Transformer with self-loops added for the analysis, but discard them in our main experiments as it leads to better performance (for reviewer sh4F)
- Table 2: Corrected the caption to show that the regularizer helps in reducing computational cost while retaining performance (for reviewer Nfpt)
- Lines 285-292 + Table 3: Added a paragraph that discusses the efficiency aspect of SBM-Transformer, with results on relative FLOP counts and peak memory usage (for reviewers ukVv, sh4F, Nfpt)
- Section 5.3: Added a new experimental section with results on the GLUE benchmark (for reviewer ukVv)
- Lines 327-336: Revised the statement regarding the main strengths of SBM-Transformer (for reviewer sh4F)

---

### Comment · Area_Chair_3gp4 · 2022-08-07
**Discussion period**

Thank you to all the reviewers for the great effort in reviewing the paper and the authors for the responses.

As in the discussion period, I want to ensure that reviewers have read the authors' responses and engage with the authors if needed.

If you haven't done this, could you please take a moment to read through the authors' responses, update the reviews to indicate that you have read the authors' responses, or communicate with the authors if needed? You can also share in private conversations with the reviewing team.

Please continue to share your thoughts. Thank you!

---

### Meta-Review · Area_Chair_3gp4 · 2022-08-28

**Recommendation:** Accept
**Confidence:** Less certain

**Metareview:**

This paper focuses on sparse attention modules for improving the computational cost of the transformer. The authors propose to adaptively learn the sparsity of the attention module through modeling the mask matrix as a bipartite graph generated by the stochastic block model (SBM).

While there are some concerns, such as the method not being well suited for large GPU clusters, overall, all the reviewers find the proposed method interesting and novel, so I recommend accept. But the authors are advised to revise the presentation of the SBM to improve the accessibility and readability and incorporate other reviewers' suggestions.

**Award:**

No

---

### Decision · Program_Chairs · 2022-09-14

Accept